# Scalable and Effective Generative Information Retrieval

## ABSTRACT

Recent research has shown that transformer networks can be used as differentiable search indexes by representing each document as a sequences of document ID tokens. These *generative retrieval* models cast the retrieval problem to a document ID generation problem for each given query. Despite their elegant design, existing generative retrieval models only perform well on artificially-constructed and small-scale collections. This has led to serious skepticism in the research community on their real-world impact. This paper represents an important milestone in generative retrieval research by showing, for the first time, that generative retrieval models can be trained to perform effectively on large-scale standard retrieval benchmarks. For doing so, we propose RIPOR– an optimization framework for generative retrieval that can be adopted by any encoder-decoder architecture. RIPOR is designed based on two often-overlooked fundamental design considerations in generative retrieval. First, given the sequential decoding nature of document ID generation, assigning accurate relevance scores to documents based on the whole document ID sequence is not sufficient. To address this issue, RIPOR introduces a novel prefix-oriented ranking optimization algorithm. Second, initial document IDs should be constructed based on *relevance* associations between queries and documents, instead of the syntactic and semantic information in the documents. RIPOR addresses this issue using a relevance-based document ID construction approach that quantizes relevance-based representations learned for documents. Evaluation on MSMARCO and TREC Deep Learning Track reveals that RIPOR surpasses state-of-the-art generative retrieval models by a large margin (e.g., 30.5% MRR improvements on MS MARCO Dev Set), and perform better on par with popular dense retrieval models.

## KEYWORDS

Generative retrieval, neural ranking models, ranking optimization

**ACM Reference Format:**
Anonymous Author(s). 2024. Scalable and Effective Generative Information Retrieval. In *Proceedings of The 2024 ACM Web Conference (WWW '24)*. ACM, New York, NY, USA, 11 pages. https://doi.org/XXXXXXX.XXXXXXX

## 1 INTRODUCTION

Pre-trained foundation models have been employed in the development of a range of retrieval models, including those that re-weight terms within queries and documents for sparse retrieval [12, 13], cross-encoder re-ranking models [31], and dual-encoder retrieval

models [19, 47, 50, 51]. Recently, Tay et al. [42] proposed an elegant and innovative approach to information retrieval (IR) by leveraging pre-trained encoder-decoder models as differentiable search indexes (DSI). This has led to the development of a few *generative retrieval* models in the past year, such as NCI [46], DSI-QG [54], and DSI++ [29]. In these models, each document ID is a unique sequence of special document ID tokens and they are often generated autoregressively using a constrained beam search algorithm [5] for each given query.

A distinct advantages of generative retrieval over existing retrieval models includes obviating the need to retrieve based on the external memory by encapsulating collection information within the model's parameters. This design promotes end-to-end training, making it seamless to integrate with existing foundation model (e.g., GPT-4) workflows for various tasks that benefit from retrieval, such as open-domain question-answering, fact verification, and conversational search [15, 43]. However, despite the theoretical appeal, prior work has only been able to demonstrate the empirical success of generative retrieval models on small-scale (and often artificially-constructed) document collections. For example, a simple term matching model, such as BM25, achieves 300% higher MRR than DSI [42] on MSMARCO, and this gap can be reduced to 76% after data augmentation through query generation [54].[1] These observations have recently led to serious skepticism in the research community on the real-world impact of generative retrieval models [33].

We argue that the poor performance of generative retrieval models is a result of two often-overlooked design considerations that are vital to their efficacy. The first pertains to the sequential nature of the beam search algorithm employed during document ID decoding. For each given query, beam search [41] sustains a top $k$ candidate list at each decoding step based on the cumulative scores of the already-decoded tokens (i.e., prefix of document IDs). In order to successfully generate the document ID of relevant documents, every document ID prefix of the relevant documents should be among the top $k$ candidate list in beam search decoding. This essential aspect is not considered by existing generative retrieval models. To address this issue, we advocate a prefix-oriented ranking optimization method, introducing a novel margin-based pairwise loss function that guides the model towards producing higher relevance score for every prefix of the relevant document IDs verses non-relevant document ID. This method also incorporates progressive training, gradually refining the model's prediction from the shortest prefix to the full-length document ID. Multi-objective progressive learning is applied to prevent the model from forgetting to emphasize on document ID prefixes.

Secondly, existing methods do not consider *relevance* information in constructing the initial document IDs. They instead use syntactic and semantic information in the documents, represented by pre-trained BERT [11] or sentence-T5 [30] to form the initial document IDs using hierarchical clustering [42, 46], ngrams [3],

---

[1] For more information, see Table 1. Similar observations have been made in [46].

or approximation methods [36, 52]. However, as demonstrated by relevance-based word embedding [48], relevance information cannot simply be captured by models trained with syntactic, semantic, and proximity based objectives. And since generative retrieval models conduct optimization with fixed document IDs, inappropriate initial construction of document IDs leads to a bottleneck inherently influencing the effectiveness of generative retrieval models. We address this issue by presenting a novel pre-training phase for initial document ID construction. Here, we transform the encoder-decoder generative retrieval model to a special dense retrieval model, with a relevance-based objective trained on the target task. The trained document representations are then decomposed into multiple vectors using residual quantization (RQ) [1, 6] that has proven to be a successful approximation for relevance-based representations.

We conduct experiments on standard large-scale information retrieval benchmarks, including MSAMRCO [4] and TREC 2019-20 Deep Learning Track data [9, 10], The retrieval collection consists of 8.8 million passages. Our approach achieves substantial improvements compared to state-of-the-art generative retrieval models in all settings. For example, our RIPOR framework[2] outperforms the best performing generative retrieval model by 30.5% in terms of MRR@10 on MSMARCO. In most settings, our model also shows better performance compared to popular dense retrieval models, such as DPR [19], ANCE [47], MarginMSE [16], and TAS-B [17]. Therefore, this paper sets an important milestone in generative retrieval research by demonstrating, for the first time, the feasibility of developing generative retrieval models that perform effectively at scale, and paving the path towards their implementation in real-world applications. To foster research in this area, we open-source our implementation and release the learned model parameters.[3]

## 2 INTRODUCTION TO GENERATIVE IR

In generative document retrieval, each document is symbolized by a unique identifier, known as document ID or *DocID* for short. Pre-trained encoder-decoder models, such as T5 [35], are employed to generate a list of document IDs in response to a given query. Let $M$ represent a generative retrieval model that represents a document $d$ using the document ID $c_d = [c_1^d, c_2^d, \ldots, c_L^d]$ of length $L$. Various methods are applied to the DocID construction [3, 42, 52]. For instance, DSI [42] employs the hierarchical k-means over the document embeddings obtained from the pre-trained BERT model [11]. Once the tree is built, each root-to-leaf path is used as a unique document ID.

As depicted in Figure 1, $M$ is trained to generate document IDs autoregressively for any given query $q$, meaning that it generates each DocID token $c_i^d$ conditioned on previously generated tokens, denoted by $c_{<i}^d$. Therefore, the model generates a conditional hidden representation for the $i^{\text{th}}$ DocID token as follows:

$$\mathbf{h}_i^d = \text{Decoder}(c_{<i}^d; \text{Encoder}(q)) \in \mathbb{R}^D.$$

where $c_{<i}^d = [c_1^d, c_2^d, \ldots, c_{i-1}^d]$ is fed to the decoder as its input and the encoded query vector is used to compute cross-attentions to the decoder. In generative retrieval, each DocID token is associated with a $D$-dimensional representation. Let $\mathbf{E}_i \in \mathbb{R}^{V \times D}$ denotes a

[2]RIPOR stands for relevance-based identifiers for prefix-oriented ranking.
[3]http://anonymized_url/

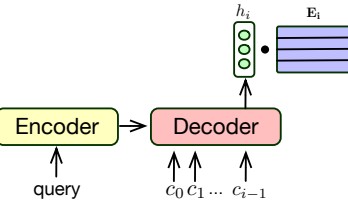

**Figure 1: An illustration of generative retrieval models.**

token embedding table for each position $i$ in the DocID sequence, where $V$ is the vocabulary size for DocID tokens, i.e., the number of distinct tokens for representing document IDs. Therefore, the representation associated with each DocID token $c_i^d$ is represented as $\mathbf{E}_i[c_i^d] \in \mathbb{R}^D$. Note that the DocID token embedding matrices are distinct, thus $\mathbf{E}_i \neq \mathbf{E}_j : \forall i \neq j$.

Inspired by seq2seq models[8, 32, 35], existing generative retrieval models estimate relevance scores based on `log-conditional probability` as follows:

$$S(q, c_d) = \log p([c_1^d, c_2^d, \ldots, c_L^d]|q)$$

$$= \sum_{i=1}^{L} \log p(c_i^d | q, c_{<i}^d)$$

$$= \sum_{i=1}^{L} \left[ \text{LogSoftmax}(\mathbf{E_i} \cdot \mathbf{h_i^d})[c_i^d] \right]$$

where $S(q, c_d)$ denotes the scoring function for a query-document pair. In this paper, we instead adopt a `conditional logit` approach, due to its less expensive computation cost and better alignment with our margin-based pairwise loss. We will further elaborate this choice in Section 3.1. This approach is inspired by dense retrieval models that use dot product similarity between query and document representations, and computes dot product similarity between the token embedding vectors corresponding to the DocID and the hidden vectors learned for each decoding position given the query and past decodings. In more detail, this approach can be formulated as follows:

$$S(q, c_d) = \text{concat}(\mathbf{E_1}[c_1^d], \ldots, \mathbf{E_L}[c_L^d]) \cdot \text{concat}(\mathbf{h}_1^d, \ldots, \mathbf{h}_L^d)$$

$$= \sum_{i=1}^{L} \mathbf{E_i}[c_i^d] \cdot \mathbf{h_i^d}.$$

Employing these scoring functions, generative retrieval models produce a ranked list of document using *beam search* with constrained decoding [5], where the top $K$ valid DocIDs are generated according to the scoring function. Each of the DocIDs is then mapped back to its original document. This results in a ranked list of $K$ documents.

## 3 METHODOLOGY

This paper proposes RIPOR, a generic framework for document ID construction and prefix-oriented ranking optimization that can be applied to any encoder-decoder architecture and enhances the performance of generative retrieval models. The high-level overview of the RIPOR framework is illustrated in Figure 2. Initially, the generative model $M$ is viewed as a dense encoder and is subjected

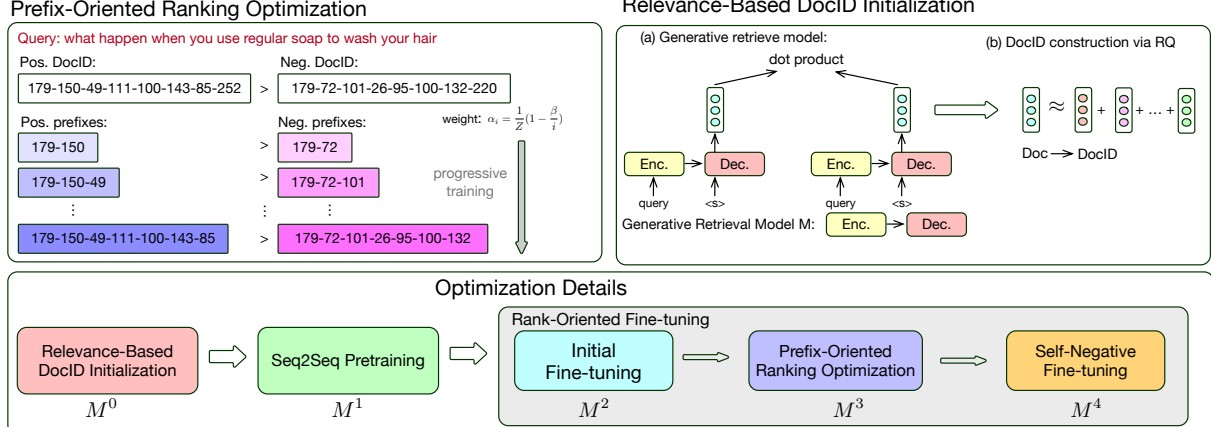

**Figure 2: The overview of the RIPOR framework**

to fine-tuning with a relevance-based objective. Upon training, RIPOR employs Residual Quantization (RQ) [6] to derive a unique identifier for each document. Subsequently, following Pradeep et al. [33], Wang et al. [46], Zhuang et al. [54], we leverage a seq2seq pre-training approach for pre-training the model using pseudo queries generated from the documents. Next, we introduce a novel rank-oriented fine-tuning procedure for refining the parameters of model $M$. In the next two sections, we elucidate the motivations and methodologies behind the two major novel components in RIPOR: prefix-oriented ranking optimization and relevance-based document ID construction. A detailed description of the entire optimization pipeline in presented in Section 3.3.

## 3.1 Prefix-Oriented Ranking Optimization

State-of-the-art generative retrieval models, such as LTRGR [23], adopt a learning-to-rank loss for optimization. The objective is to ensure that $S(q, c_{d^+}) > S(q, c_{d^-})$ for a training triplet of query $q$, relevant document $d^+$ and irrelevant document $d^-$. We posit that this modeling is not optimal. A primary oversight is the intrinsic nature of beam search that *sequentially* decodes document ID tokens from left to right. Solely focusing on pairwise ranking for a full-length document ID does not guarantee that relevant documents can survive the beam search eliminations in earlier decoding steps. Therefore, we aim at developing a model that produce accurate scoring at every decoding step. Formally, we desire to satisfy the following criterion: $S^i_{\text{prefix}}(q, c_{d^+}) \geq S^i_{\text{prefix}}(q, c_{d^-}), \quad \forall i \in [1, L]$, where $S^i_{\text{prefix}}(q, d)$ denotes the relevance score produced by the generative retrieval model for the first $i$ tokens in the document ID: $[c_1^d, c_2^d, \ldots, c_i^d]$.

***Margin Decomposed Pairwise Loss***. Taking inspiration from MarginMSE [16], a pairwise loss for knowledge distillation as follows:

$$\mathcal{L}(q, d^+, d^-) = \left( S(q, d^+) - S(q, d^-) - T_{(q,d^+,d^-)} \right)^2,$$

where $T_{(q,d^+,d^-)}$ denotes the golden margin, commonly predicted by a teacher model derived from a cross-encoder [31]. Prior research [16, 50] reveals that this loss function often outperforms other pairwise losses [47] by addressing data sparsity issues in large-scale retrieval benchmark [34], utilizing pseudo-labels for unlabeled query-document pairs.

For generative retrieval, we extend the MarginMSE loss by modeling pairwise ranking between prefixes of $c_{d^+}$ and $c_{d^-}$ for each decoding step $i$:

$$\mathcal{L}^i_{\text{rank}}(q, c_{d^+}, c_{d^-}) = \left( S^i_{\text{prefix}}(q, c_{d^+}) - S^i_{\text{prefix}}(q, c_{d^-}) - \alpha_i T_{(q,d^+,d^-)} \right)^2.$$

Here, at each step $i$ we re-weight the golden margin by multiplying with $\alpha_i$, which is a weight we assign to each prefix position. The reason for this decision is that we emphasize on the early decoding steps of the document IDs. With this motivation, $\alpha_i$ should be a monotonically increasing concave function w.r.t. $i$. Formally, $\alpha_i$ values should satisfy the following constraint: $\alpha_i - \alpha_{i-1} \geq \alpha_{i+1} - \alpha_i$ for every $i$. In our experiments, we use $\alpha_i = \frac{1}{Z}(1 - \frac{\beta}{i})$, where $Z = 1 - \frac{\beta}{L}$ is a normalization factor and $\beta$ is a constant hyper-parameter. We leave the exploration of other concave functions to future work. For efficiency reasons, we only do prefix-oriented optimization for $i = 4, 8, 16, 32$ and thus set $\beta = 2$. This concave formulation of $\alpha_i$ emphasizes larger sub-margins in early steps, ensuring for any query $q$ that $S^i_{\text{prefix}}(q, c_{d^+})$ surpasses $S^i_{\text{prefix}}(q, c_{d^-})$. Moreover, as $\alpha_L = 1$, the predicted margin for full-length DocID sequences aligns with the real margin, maintaining the fidelity of ranking knowledge.

***Progressive Training***. To better learn representations aligned with the left-to-right decoding characteristic of the beam search, we draw inspiration from curriculum learning [2, 26, 28, 50] and implement a progressive training strategy. The training process is initialized with the shortest prefix. This allows the model to first focus on basic sequence representations and build adequate capacity for the subsequent stages. As the training advances, the scope is systematically extended to the longer prefixes, culminating in training on the full-length sequence with length $L$.

During training on longer prefixes, we empirically found that the model tends to overlook previously acquired knowledge related to shorter prefixes. To mitigate this catastrophic forgetting issue, we employ multi-objective learning at each time step to ensure the retention of knowledge acquired in earlier stages. Given the training data $\mathcal{D} = \{(q_j, d_j^+, d_j^-, T_{(q_j, d_j^+, d_j^-)})\}_{j=1}^{|\mathcal{D}|}$, we use the following multi-objective loss function:

$$\sum_{(q, d_j^+, d_j^-) \in \mathcal{D}} \big( \underbrace{\mathcal{L}_{\text{rank}}^i(q, d_j^+, d_j^-)}_{(1)} + \underbrace{\sum_{k=1}^{i-1} \mathcal{L}_{\text{rank}}^k(q, d_j^+, d_j^-)}_{(2)} \big)$$

In this loss function, term (1) is responsible for acquiring the pairwise rankings specific to the current step $i$, while term (2) ensures the model retains the ranking knowledge from previous prefixes. As mentioned earlier, for efficiency reasons, without loss of generality, we only repeat this training process for $i = 4, 8, 16, 32$.

## 3.2 Relevance-Based DocID Construction

Generative retrieval models predominantly adopt a two-step optimization approach. First, they initialize the document IDs by employing various methods such as hierarchical k-means [42, 46] or discriminative textual descriptions extracted from documents [3, 23, 24]. In the subsequent step, they optimize the model leveraging either cross-entropy loss [3, 42] or learning-to-rank loss [23], with fixed DocIDs obtained in the first step. Given that the DocIDs remain immutable in this phase, they potentially become a significant bottleneck, influencing the overall efficacy of generative retrieval models.

We argue that the design of DocIDs is crucial in two specific ways: First, it must ensure the documents with inherent similarity possess correspondingly similar DocIDs. Second, due to the characteristics of beam search for decoding in generative retrieval, these DocIDs should encapsulate a hierarchical structure. Notably, the conception of similarity in this context is nuanced; it is tied intricately to specific queries and deviates from standard linguistic similarities observed in natural language processing. Addressing these challenges, we introduce a relevance-based method for initializing DocIDs. This approach is crafted to encapsulate both the query-document relevance nuances and the necessary hierarchical structure, ensuring effective performance in generative retrieval tasks.

***Generative retrieval model as dense encoder***. To capture the relevance-based similarities among documents, we design an optimization process inspired by dense retrieval models, but by utilizing the encoder-decoder architecture in $M$. Specifically, we input document content into the encoder and a special start token as input to the decoder. The document representation is then derived from the first contextualized output embedding of the decoder:

$$\mathbf{d} = \text{Decoder}(s_0; \text{Encoder}(d)) \in \mathbb{R}^D.$$

Where $s_0$ is the start token. Adopting a similar approach for queries, we determine their representations. To optimize model $M$, we employ the MarginMSE loss [16] with multi-stage negative sampling introduced in Sec 3.3.1 in details.

***Residual Quantization***. Hierarchical k-means, which is used in [33, 42, 46, 54] for document ID construction, does not explicitly minimize the distortion error between original and approximated representations. As highlighted by Ge et al. [14], there is a notable inverse correlation between information retrieval metrics like MAP and the distortion error, particularly for large-scale datasets. Motivated by this observation, we adopt quantization-based techniques [1, 6, 14, 45] explicitly designed to minimize this distortion error. Among a myriad of quantization algorithms, we select Residual Quantization (RQ) [1, 6] due to its inherent advantages. Specifically, (1) its recursive procedure captures the hierarchical document structure, aligning with the beam search strategy inherent to generative retrieval, and (2) compared to methods like product quantization (PQ) [14, 45], it requires a shorter length of DocID to achieve a strong performance, leading to memory and time savings during inference. Using $M$ as our dense encoder, we calculate the representation $\mathbf{d}$ for each document $d$. Subsequently, employing RQ, we optimize the token embedding table $\{\mathbf{E}_i\}_{i=1}^L$ to determine the optimal DocID $c_d = [c_1^d, \ldots, c_L^d]$ for every document $d$. Upon optimization, each $\mathbf{d}$ can be approximated using a sequence of token embeddings as:

$$\mathbf{d} \approx \sum_{i=1}^L \mathbf{E}_i[c_i^d].$$

The trained model $M$ alongside the embedding tables $\{\mathbf{E}_i\}_{i=1}^L$ will serve as the initial weights for subsequent optimization phases within generative retrieval.

## 3.3 Optimization Details

Our optimization process can be delineated into three distinct phases: (1) DocID initialization (2) Seq2seq Pre-training, and (3) Rank-oriented Fine-tuning.

*3.3.1 DocID Initialization.* As described in Section 3.2, we treat $M$ as a dense encoder. To optimize the dense encoder $M$, we use the recent advance of multi-stage training strategy [47]. Here's the tailed steps of the multi-stage training: In the initial stage, we use BM25 [38] to sample the top $K$ (We choose $K = 100$ in our work) documents for each query and train the model using the MarginMSE [16] loss function. Once the model is trained, we obtain the dense representation $\mathbf{d}$ from our model $M$ for each document and store them in an index. For each query $q$, we apply nearest neighbor search to retrieve the top $K$ documents. Then, we train the model using the same loss function with the retrieved documents. After training, we then apply residual quantization (RQ) to obtain the DocID for each document. The trained model is denoted as $M^0$, and the embedding tables $\{\mathbf{E}_i\}_{i=1}^L$ will be used as the initial weights for the next phase.

*3.3.2 Seq2seq Pre-training.* To equip our model $M$ with a comprehensive understanding of the corpus, we incorporate a seq2seq pre-training phase. Instead of using the document $d$ as input and predicting its corresponding semantic tokens $[c_1^d, \ldots, c_L^d]$, we align with prior work [46] and utilize pseudo queries associated with each document as input proxies for DocID prediction. Specifically, by leveraging the doc2query model [7], we generate $N_{pseudo}$ pseudo queries for every document. We then optimize the model using

a cross-entropy loss, with the tokens from the relevant DocIDs serving as the ground-truth labels. We denote the trained model in this phase as $M^1$.

### 3.3.3 Rank-oriented Fine-tuning.
To optimize our model, we leverage the pairwise loss as described in Sec 3.1. Literature suggests the pivotal roles of negative sampling [47] and the quality of the supervision signal [16, 17, 50] in enhancing the performance of ranking models. Following this, we incorporate a multi-stage training strategy [47, 51] to incrementally enhance the model's capacity and extract improved negatives for subsequent stages.

*Initial Fine-tuning:* This stage is primarily geared towards further preparing the generative retrieval model for the ranking task and sourcing high-quality negative samples for ensuing stages. Utilizing the model $M^0$ from Sec 3.3.1 as a dense encoder, we index each document via its embedded representation. We apply the Nearest Neighborhood search to retrieve the top 100 documents. The training data $\mathcal{D}^R$ can be constructed based on the negative samples and ground-truth query-document positive pairs. Unlike our approach in subsequent stages, we directly utilize the full-sequence ranking loss $\mathcal{L}_{rank}^L$. Starting from $M^1$ as an initial model, after training, the model is represented as $M^2$. This is intentional, as the primary objective here is to curate quality negative samples for later stages rather than perfecting the model.

*Prefix-Oriented Ranking Optimization:* Given a query $q$, we deploy beam search on the model $M^2$ to retrieve the top 100 DocIDs, each of which is mapped back to corresponding documents. The documents serve as an augmented source of negative samples, and we subsequently construct a training set $\mathcal{D}^B$ in a manner analogous to the previous section. The comprehensive training set for this stage consolidates data both from the Nearest Neighborhood Search and Beam Search, represented as $\mathcal{D} = \mathcal{D}^R \cup \mathcal{D}^B$. To optimize the model, we utilize the progressive training described in Section 3.1. For each optimization step $i$, we employ the multi-objective loss function described in Section 3.1. After training, the model is denoted as $M^3$.

*Self-Negative Fine-tuning:* To enhance the model's effectiveness, we employ beam search on the most recently optimized model $M^3$ to establish a training dataset $\mathcal{D}_{self}^B$. Then the model is trained on the same multi-objective loss function in the full-length setting ($i = L$), and denoted as $M^4$.

## 4 EXPERIMENTS

### 4.1 Experiments Settings

#### 4.1.1 Dataset.
We assess our information retrieval models on the MSMARCO dataset [4], comprising 8.8M passages and 532K training queries with shallow annotations (averaging about 1.1 relevant passages per query). We evaluate our models using three datasets: (1) MSMARCO-Dev, with 7K queries and shallow annotations, (2) TREC DL 2019: the passage retrieval dataset used in 2019 TREC Deep Learning Track [9] with 43 queries and (3) TREC DL 2020: the passage retrieval dataset of TREC Deep Learning Track 2020 [10] with 54 queries. For evaluation, we report recall@10 for all datasets, as well as the official metric MRR@10 for the MSMARCO-Dev set and NDCG@10 for the TREC DL 2019 and 2020.

#### 4.1.2 Implementation Details.
We employ the pre-trained T5-base [35] as the backbone for our generative retrieval model. For DocID initialization, we adopt the residual quantization (RQ) implementation from Faiss [18]. The length of DocID $L$ is 32 and the table size $V$ is 256. For Seq2seq pre-training, the doc2query model [7] with t5-large as the backbone generates 10 pseudo queries for each document. For progressive training, we sample 4 prefixes with lengths 4, 8, 16, 32. The optimization is done using Adam [21], featuring linear scheduling and a warmup ratio of 4.5% of total learning steps. For DocID initialization and rank-oriented fine-tuning phases, we set the learning rate as 0.0001 with 120 epochs and batch size of 64 For Seq2seq pre-training, we set the learning rate as 0.001 with 250,000 steps and batch size of 256 We conducted all the experiments using 8 A100 GPUs.

#### 4.1.3 Baselines.
We select a wide range of document retrieval models from generative retrieval to dense retrieval as the baselines for comparison:

- **DSI** [42]: DSI is one of the earliest generative retrieval models that apply the hierarchical k-means over document representations obtained from pre-trained BERT for DocID construction. The model utilizes cross-entropy loss for fine-tuning on the retrieval task.
- **DSI-QG** [54]: DSI-QG generates pseudo queries for each document and uses them as the augmented data for training.
- **NCI-QG** [46]: NCI-QG invents a prefix-aware weight-adaptive decoder architecture to capture position information of document identifiers, and like DSI-QG, uses the doc2query model for data augmentation.
- **SEAL** [3]: SEAL employs document n-grams as identifiers, applying the FM-index to ensure valid document identifiers are decoded in response to specific queries.
- **MINDER** [24]: An extension of SEAL, MINDER constructs document identifiers from multiple document views, such as titles, pseudo-queries, and n-grams.
- **LTRGR** [23]: LTRGR utilizes multi-view document identifiers, akin to MINDER, but shifts the loss function to a pairwise-based learning-to-rank algorithm.
- **BM25** [38]: the simple yet effective bag-of-word retrieval model that uses term-frequency, inverse document frequency, and document length for computing the relevant scores
- **DPR** [19]: DPR is a dual-encoder based dense retrieval models. It incorporates the in-batch negative and BM25 negatives for training.
- **ANCE** [47]: ANCE selects hard training negatives from the entire corpus by using an asynchronously updated ANN index.
- **MarginMSE** [16]: MarginMSE develops a distinctive loss function based on the konwledge distillation. It aims to minimize the discrepancy between the predicted margin from dense retrieval models and the golden margin from the teacher model.
- **TAS-B** [17]: Building upon MarginMSE, TAS-B designs a topic-aware sampling algorithm to enhance the model's effectiveness.

**Table 1: Experimental results on MSMARCO and TREC Deep Learning Track Data. Highest generative retrieval performances are boldfaced. Superscript ∗ denotes statistically significant improvement compared to all generative retrieval baselines. Superscripts △ and ▽ denote significantly higher and lower performance compared to RIPOR. (t-test with Bonferroni correction, p_value < 0.01). For dense retrieval models, HNSW [27] index is used for ANN search.**

| Model | MSMARCO Dev | | TREC DL 2019 | | TREC DL 2020 | |
|---|---|---|---|---|---|---|
| | MRR@10 | Recall@10 | NDCG@10 | Recall@10 | NDCG@10 | Recall@10 |
| **Generative Retrieval** | | | | | | |
| DSI | .045 | .138 | .163 | .076 | .150 | .070 |
| DSI-QG | .105 | .292 | .320 | .138 | .328 | .120 |
| NCI-QG | .153 | .352 | .403 | .167 | .394 | .159 |
| SEAL | .127 | - | - | - | - | - |
| MINDER | .186 | .383 | .506 | .201 | .392 | .144 |
| LTRGR | .255 | .531 | - | - | - | - |
| **RIPOR (ours)** | **.333**∗ | **.562**∗ | **.628**∗ | **.205**∗ | **.631**∗ | **.191**∗ |
| **Sparse and Dense Retrieval Models (For Reference)** | | | | | | |
| BM25 | .185▽ | .381▽ | .512▽ | .178▽ | .477▽ | .164▽ |
| DPR | .287▽ | .539▽ | .588▽ | .195▽ | .581▽ | .182▽ |
| ANCE | .301▽ | .545▽ | .600▽ | .262▽ | .587▽ | .174▽ |
| MarginMSE | .312▽ | .552▽ | .634△ | .250△ | .614▽ | .193 |
| TAS-B | .323▽ | .557▽ | .629 | .200 | .633 | .227△ |

**Table 2: Ablation study results on MSMARCO Dev. Superscript ▽ denotes significantly lower performance compared to RIPOR (t-test with Bonferroni correction, p_value < 0.01).**

| | MRR@10 | Recall@10 |
|---|---|---|
| -. RIPOR | .333 | .562 |
| 1. w/o prefix optimization | .280▽ | .475▽ |
| 2. w/o multi-objective learning | .317▽ | .532▽ |
| 3. w/o self-neg. fine-tuning | .325▽ | .543▽ |
| 4. w/o seq2seq pre-training | .319▽ | .539▽ |
| 5. replace with sentence t5 | .192▽ | .287▽ |
| 6. replace with PQ | .112▽ | .155▽ |

**Table 3: The retrieval performance for various DocID combinations on MSMARCO Dev set.**

| $L \times V$ | MRR@10 | Recall@10 | Extra Param.(M) |
|---|---|---|---|
| $32 \times 256$ | .333 | .562 | 6.29 |
| $16 \times 512$ | .307 | .520 | 6.29 |
| $8 \times 1024$ | .306 | .535 | 6.29 |
| $4 \times 2048$ | .273 | .493 | 6.29 |
| $16 \times 1024$ | .324 | .554 | 12.58 |
| $8 \times 2048$ | .319 | .550 | 12.58 |
| $4 \times 4096$ | .291 | .528 | 12.58 |

## 4.2 Experiment Results

*4.2.1* ***Main Results***. We report the performance of RIPOR and other baselines MSMARCO in Table 1. First, most generative retrieval models, including DSI, DSI-QG, NCI-QG, SEAL, and MINDER, consistently lag behind BM25 across all three evaluation sets. In contrast, the LTRGR model, which incorporates a learning-to-rank algorithm, manages to surpass BM25. These observations underscore the importance of integrating learning-to-rank methodologies when designing generative retrieval models. Second, our proposed RIPOR consistently outperforms other generative retrieval baselines, demonstrating a significant advantage. Notably, when compared to the top-performing baseline LTRGR, RIPOR achieves a 30.5% improvement in MRR@10 on the MSMARCO Dev set and a remarkable 94% enhancement in NDCG@10 on the TREC-20 test set. Third, our RIPOR can obtain comparable results to state-of-the-art dense retrieval models, particularly in precision-oriented metrics. For instance, compared to ANCE, our model achieves a 16% improvement in terms of MRR@10 on MSMARCO Dev and

a 6.6% improvement on the two TREC DL evaluation sets in total NDCG@10[4]. Additionally, we provide the experimental results on the small-scale dataset MSMARCO-1M, in line with previous work [33]. These results can be found in Appendix Table 4.

*4.2.2* ***Ablation Studies***. We conduct a thorough ablation studies on the MSMARCO dataset to investigate the impact of each component in RIPOR. We report our study in Table 2.

Beginning with Row 1, we can see the significance of incorporating prefix-oriented ranking optimization. The absence of this optimization results in a pronounced 19% degradation in MRR@10. Without employing the optimization approach, the model fails to explicitly ensure that every prefix of relevant DocIDs receives higher scores than those of relevant DocIDs in response to a query. This increases the risk of discarding these relevant DocIDs in the early steps of beam search, which, in turn, negatively impacts information performance.

---

[4]HNSW index might slightly impact the performance compared to DR models using a flat index [27, 47]

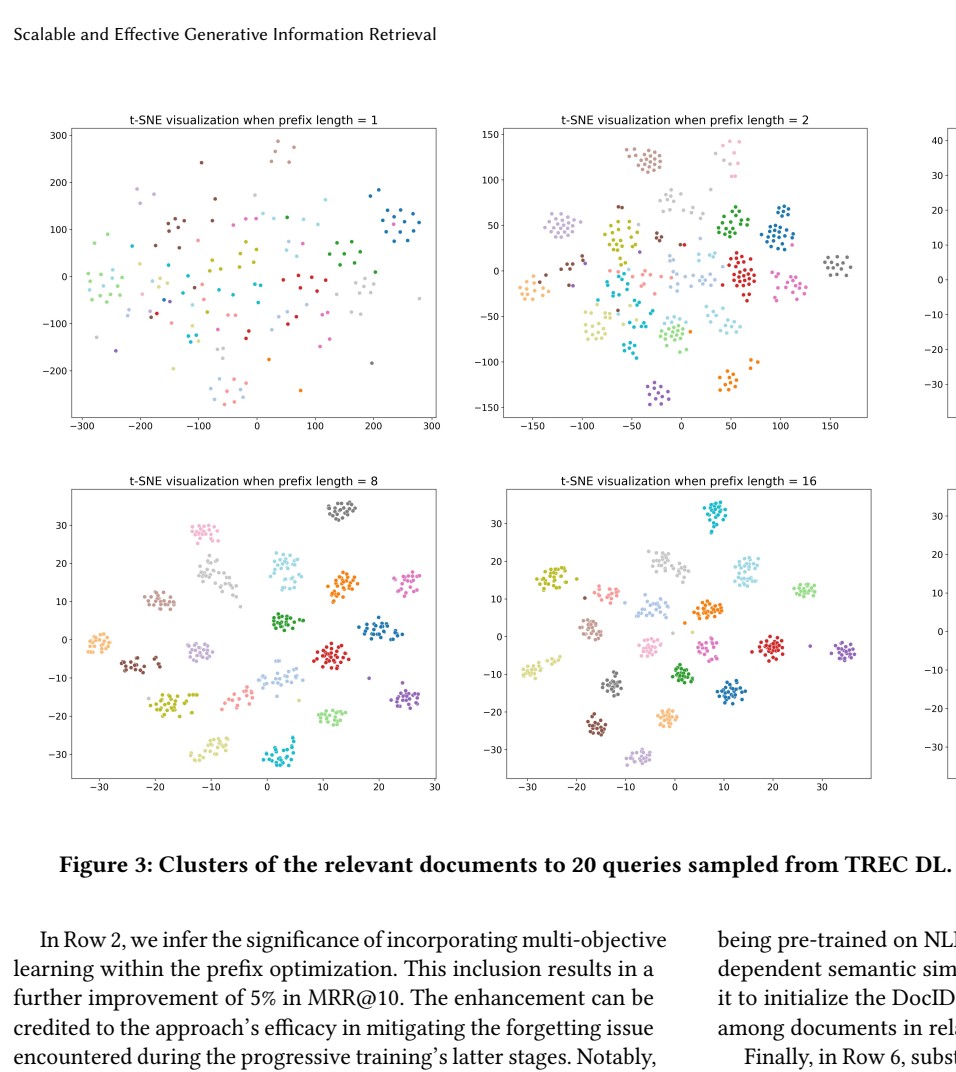

**Figure 3: Clusters of the relevant documents to 20 queries sampled from TREC DL. The color indicates the query ID.**

In Row 2, we infer the significance of incorporating multi-objective learning within the prefix optimization. This inclusion results in a further improvement of 5% in MRR@10. The enhancement can be credited to the approach's efficacy in mitigating the forgetting issue encountered during the progressive training's latter stages. Notably, this methodology introduces only a minimal addition to the loss computation, ensuring that there is no increase in computational overhead during training.

Row 3 reports the results for RIPOR when self-negative fine-tuning is not used in the final training stage. Incorporating this strategy yields a 2.5% enhancement in MRR@10 and a 3.5% boost in Recall@10. This improvement stems primarily from the fact that hard negative samples would increae the efficacy of the retrieval model as shown in previous dense retrieval models[47]. By strategically leveraging these hard negative samples, we bolster the model's capability, ensuring relevant DocIDs consistently be ranked higher than potential high-scoring hard negatives, which subsequently elevates the model's overall effectiveness.

From Row 4, we note that by integrating seq2seq pre-training, RIPOR achieves a 4% improvement in MRR@10. This method allows the model to encapsulate document information across the entire corpus, mirroring the indexing phase in dense retrieval models, and subsequently driving the observed performance improvement.

From Row 5, when we treat the generative retrieval model as a dense encoder and instead use the sentence-T5 [30] to derive the hidden representation for each document, a substantial performance degradation would happen, with a 73% drop in MRR@10, for instance. The rationale behind this decline is that sentence-T5,

being pre-trained on NLP tasks, is not optimized to discern query-dependent semantic similarities between documents. Leveraging it to initialize the DocIDs disrupts the inherent semantic linkages among documents in relation to queries.

Finally, in Row 6, substituting RQ with PQ results in a substantial performance decline, evidenced by a 197% decrease in MRR@10. While PQ is recognized as a potent quantization algorithm in the dense retrieval domain, our results suggest its unsuitability for generative retrieval. This limitation may stem from PQ's inability to encapsulate the hierarchical structure among documents, an attribute that has been shown to be crucial in generative retrieval, especially when employing beam search.

### 4.3 Analysis and Discussion

#### 4.3.1 *The impact of DocID combination*. The configuration of the Document Identifier (DocID), specifically its length $L$ and vocabulary size $V$, influences the effectiveness of model $M$. We examine this relationship by evaluating various performance metrics on the MSMARCO Dev set, as detailed in Table 3. Firstly, when holding the extra parameters constant (quantified by $L \times V \times D$), we observe that an elongation in DocID length $L$ corresponds to enhanced performance in both MRR@10 and Recall@10. Secondly, while maintaining a fixed DocID length $L$ and incrementing the vocabulary size $V$, there's a noticeable improvement in performance metrics. For instance, when $L = 16$, increasing the vocabulary size from 512 to 1024 leads to the 5.5% improvement in MRR@10.

#### 4.3.2 *The quality of document approximated representation*. In Section 3.2, we emphasized the importance of the relevance-based

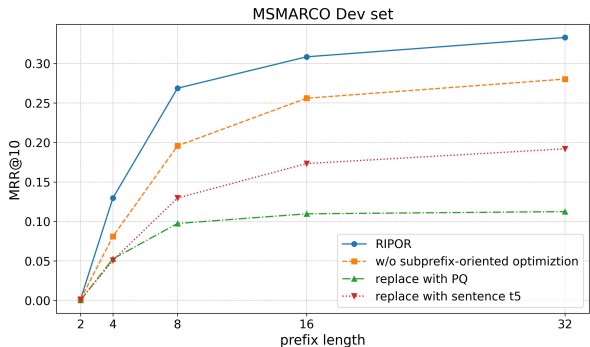

**Figure 4: The retrieval performance for different prefix lengths in MSMARO Dev.**

document similarities, in influencing model performance. To prove that our model can capture these signals. We randomly selected 20 queries from TREC DL 19 and TREC DL 20, along with their corresponding relevant documents. We utilize the approximated vector representation $\hat{\mathbf{d}} = \sum_{i=1}^{L} \mathbf{E}_i[c_i^d]$ and apply T-SNE [44] to project the approximated representations of each document into a 2D space for visualization. We studied clustering quality for different prefix lengths, specifically $L = 1, 2, 4, 8, 16$, and 32, as illustrated in Figure 3. First, when $L \geq 8$, those documents with the same relevant query are located in their corresponding cluster nicely, which indicates that our RIPOR effectively draws relevant documents nearer while distancing the irrelevant ones. Second, the clustering quality is progressively improved when $L$ increases. This might be because when $L$ increases, the distance between approximated vector $\hat{\mathbf{d}}$ and original vector $\mathbf{d}$ diminishes, enabling the approximation to capture finer-grained ranking information.

*4.3.3* ***The influence of prefix-length***. The prefix length plays a pivotal role in the RIPOR framework due to its influence on the distortion error between the original and approximated vectors. While Section 4.3.2 provides a qualitative perspective on its effects in terms of document similarities in a low-dimensional space, this section delves into its quantitative impact on retrieval performance, as depicted in Figure 4. Referring to the left figure, which displays different DocID combinations from RIPOR, several trends emerge. First, as the prefix length $L$ grows, there's a consistent improvement in performance. Second, the rate of this performance gain is more pronounced for shorter prefix lengths, since we observe that the boost is more substantial when $L \leq 8$ than when $L > 8$. Third, given an equal prefix length, variants with a larger vocabulary size tend to perform better. From the right figure, which contrasts RIPOR with three other selected variants from the ablation study in Section 4.3.3: First, excluding the prefix-oriented optimization invariably results in reduced performance. Second, the performance curve of the "replace with sentence-T5" variant emphasizes the critical role of DocID initialization. Its subpar performance suggests that not employing relevance-based DocID initialization is detrimental, rendering it less effective than the variant excluding prefix-oriented optimization. Third, Product Quantization (PQ) seems less compatible with generative retrieval, given its stagnant performance for $L \geq 8$. This stagnation might be due to PQ's shortcomings in capturing the hierarchical nuances among documents, subsequently impacting the benefits drawn from longer prefix lengths.

## 5  RELATED WORK

Pre-trained language models (LMs) [11, 25, 35, 40] have become foundational in the field of information retrieval (IR). The integration of these LMs into neural IR models can be broadly categorized into four main streams:

Neural Sparse Retrieval models, inspired by conventional bag-of-words approaches like TF-IDF [37] and BM25 [38], adapt BERT to re-weight subwords, thereby enhancing IR performance. To maintain the sparsity of high-dimensional vectors, they utilize L1 [49] or Flop [13] regularizers. This characteristic sparsity allows them to be incorporated into fast search frameworks based on the inverted index [39].

Re-ranking with LMs is another approach where LMs serve as re-rankers [31, 53]. By feeding a concatenated query and document, these models produce a relevance score. Despite their often superior performance, they are better suited for document re-ranking due to efficiency constraints, rather than retrieval.

Dense Retrieval models, a more recent advancement, are dual-encoder-based [16, 17, 19, 20, 34, 47, 50, 51]. Notably, they have exhibited standout performance on large-scale datasets [4, 22]. These models, typically leveraging BERT, encode each document and query into dense representations. For efficient retrieval, they employ approximated nearest neighbor search (ANN) [27, 47].

Lastly, the Generative Retrieval paradigm is an innovative approach drawing inspiration from successful generative LMs [8, 32, 35]. In this paradigm, models like T5 are treated as retrievers. Each document is mapped to a distinct sequence, often denoted as a DocID. At inference, given a specific query, a constrained beam search [42, 52] retrieves a list of the most probable DocIDs.

## 6  CONCLUSIONS AND FUTURE WORK

We introduced the RIPOR framework, designed to generalize generative retrieval models for large-scale datasets. We employ a novel prefix-oriented ranking optimization method to harness the sequential nature of DocID generation. By viewing generative retrieval as a dense encoder, we fine-tune it for the target task, and apply RQ for DocID construction. Our experimental results demonstrate that this DocID construction captures the relevance-based similarity among documents, thereby improving the effectiveness of the IR task. Looking ahead, we aim to further optimize the model's efficiency and integrate the framework into other knowledge-intensive NLP tasks, such as Open-domain QA.

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

## A  PERFORMANCE ON SMALLER-SCALE DATASET.

Following the methodology in [33], we also create a scaled-down version encompassing 1M passages. Initially, we include all passages relevant to the 532K training queries and the 7K Dev set queries, summing to 522K passages. The rest are selected at random from the main collection, totaling 1M passages. We merge the TREC DL 19 and TREC DL 20 to form the TREC DL. The results for MSMARCO Dev and TREC DL are shown in the Table 4

Table 4: The performance comparison on MSMARCO-1M. Highest performance models are boldfaced (p_value < 0.01)

| Model | MSMARCO Dev | | TREC DL | |
|---|---|---|---|---|
| | MRR@10 | Recall@10 | NDCG@10 | Recall@10 |
| BM25 | .418 | .625 | .275 | .054 |
| DSI-QG | .508 | .726 | .429 | .339 |
| NCI-QG | .511 | .720 | .441 | .340 |
| DPR | .542 | .775 | .505 | .340 |
| ANCE | .547 | .779 | .514 | .338 |
| MarginMSE | .556 | .781 | .537 | .399 |
| TAS-B | .573 | .789 | **.542** | .418 |
| RIPOR | **.580** | **.793** | .530 | **.425** |

