# OpenReview forum: "Scalable and Effective Generative Information Retrieval"
_ACM.org/TheWebConf/2024/Conference — TheWebConf24 Oral_

### Official Review · Reviewer_yxq9 · 2023-11-06

**Novelty:** 6
**Technical Quality:** 5

**Review:**

The paper proposes a pipeline of model training characteristics to make generative IR scale to MSMARCO size (~8M passages). In particular, the paper addresses a ranking-oriented optimisation. Experiments are conducted using three querysets on MSMARCO.

The contribute has 4 key contributions - each of these are ablated and shown to benefit effectiveness. I like this paper and would be happy to see it in WWW. It moves on the generative IR field to be on par with some dense retrieval models.

Significant points:
 I would however note that there are other single-rep dense retrieval models that can be as effective e.g TCT-ColBERT (see https://dl.acm.org/doi/10.1145/3477495.3531721). You might argue that TCT-ColBERT is a teacher/student model with multiple training steps, but the proposed RIPOR also has multiple training stages.

I also would like some treatment of training time in this paper. We know one aspect was trained for 250k steps, but how does training time compare to other models?

Clarifications:
- In Section 4.3.3/Figure 4(right), be clear on WHAT is being replaced, e.g. WHAT is being replaced with PQ?
- the argumentation around line 408-412 about distortion error and MAP is difficult to follow. How was kmeans being used? Is distortion error different from reconstruction error?

Negative points:
 - I would like the baseline models to have been better characterised - e.g. the definition of TAS-B is incremental over MarginMSE, and a casual reader would not understand if its single-representation dense retrieval, etc. Why not make families of baselines?
- ANCE is described as a state-of-the-art dense retrieval model. I would disagree
- I found Figure 3 to follow - I think the caption could be extended.

Minor points:
- abstract: "perform better on par" -- cannot parse.
- In Table 1, why aren't 0.629 0.633 bold?
- Reformulate: "Literature suggests"
- line 534: use full stop; line 133 ditto.
- line 402: use small "where"

**Questions:**

- Please discuss the applicability of the ANCE as a state-of-the-art dense retrieval model, and discuss comparison to other effective single-rep dense retrieval models
- Please discuss training time

**Ethics Review Description:**

N.A

**Reviewer Confidence:**

4: The reviewer is certain that the evaluation is correct and very familiar with the relevant literature

**Scope:**

4: The work is relevant to the Web and to the track, and is of broad interest to the community

---

### Official Review · Reviewer_Gndm · 2023-11-22

**Novelty:** 5
**Technical Quality:** 4

**Review:**

The paper proposes a training pipeline that can achieve good performance on the MS MARCO dataset compared to previous generative models. The proposed prefix-oriented optimization, multi-objective loss, residual quantization for docIDs are fairly effective in boosting the final performance of the trained model. The author conducts experiments in various aspects and the write up is easy to follow.

Pros:
1. The authors proposed to use the conditional logit loss instead of the traditional log-conditioned probability that better suits the ranking  use case.
2. The authors compare with a good amount of generative retrieval baselines including the latest ones.
3. The ablation study demonstrates the effectiveness of each proposed component which is informative.

Cons:
1. Lacking necessary aspects in the related work: as the major contribution of the pipeline is the training loss and the way document id is constructed. I would expect a description on the history of these lines of work in the Related Work section to clarify the scope of contribution.
2. As the training pipeline is pretty complicated with pre training for docIDs, seq2seq and three rounds of fine-tuning with hard-negative mining and fine-tuning. I would expect the authors provide more experimental results on clearing up the contribution of each stage. Please refer to question 1 and 2 below for details.

**Questions:**

1. As the proposed training procedure has three rounds of fine-tuning, it would be necessary to report the model performance after each round to clearly demonstrate the improvements from round to round. Training takes time, readers might have the restrictions or preference on only training one round or two rounds. It would help reproducing and understanding the pipeline with performance after each round reported.

2. Knowledge Distillation can greatly boost a ranking model’s performance. When comparing the performance in the tables,it would be fair to compare the distilled model with baseline models also trained with knowledge distillation such as LTRGR.   For baselines trained without knowledge distillation, such as MINDER, DPR, etc., it would be great if the author could replace marginMSE with a contrastive loss in the fine-tuning stages and compare the un-distilled model with the baselines.  Based on the description of the pipeline, training a model without knowledge distillation seems to be a feasible option.

3. More evidence on Scalability: It seems that the author markets the proposed training pipeline as scalable because it outperforms other GR baselines on the relatively large MS MARCO dataset. However, to demonstrate the scalability, MS MARCO with 8 million documents doesn’t seem to be big enough. There are other datasets such as ClueWeb that is much larger. If scalability is a contribution, then the readers would expect more experimental support from the author.

4. Besides, in the ablation for docID length, vocab size, it would be very helpful if the author could also report the time latency with respect to different setup. Time efficiency would be crucial when we scale up the corpus size.

**Reviewer Confidence:**

3: The reviewer is confident but not certain that the evaluation is correct

**Scope:**

4: The work is relevant to the Web and to the track, and is of broad interest to the community

---

### Official Review · Reviewer_Pdfo · 2023-11-22

**Novelty:** 5
**Technical Quality:** 3

**Review:**

## Pros:
- The paper tackles an important problem of scaling up generative retrieval models to large datasets, which has been a major limitation of existing methods. The paper provides useful analysis on the challenges of generative retrieval, identifying two key issues: (1) the sequential nature of document ID generation and (2) using relevance signals for document ID construction. The proposed techniques of prefix-oriented ranking and residual quantization effectively address these issues.

- Thorough experiments are conducted on standard IR benchmarks like MS MARCO and TREC DL, demonstrating the efficacy of RIPOR in scaling up generative retrieval.

- This paper is well organized and presented. Most parts of the paper are easy to understand.

## Cons:
- A major advantage of generative retrieval is that it can be trained end-to-end as the authors state in Line 75. However, the proposed RIPOR requires a multi-step complex training process and also needs the golden margin predicted by the additional teacher model (i.e., dense retrieval model **MarginMSE**), which may conflict with the motivation.

- The consideration of baselines is not comprehensive enough. There are other ways of constructing semantically structured identifiers, such as Ultron-PQ[1] and GenRet[2]. The authors should compare these strongly related methods.

- When 𝐿 is set to 32 and 𝑉 is set to 256, what is the repetition rate of DocID? The author should explain how to solve the problem of multiple documents sharing the same ID.

- If two documents share similar topics, it makes sense for them to have similar prefixes. However, in general, a document may contain different topics. The authors need to explain how to deal with this case when generating prefixes for DocIDs. Some works have proposed designing multiple DocIDs to represent a document [3].

- I find the related work section less convincing. Only two works on generative retrieval are introduced in Line 914 (Section Related Work). Several noteworthy works have been published in reputable conferences such as SIGIR [4,5], CIKM [6,7], NeurIPS [2] and ACL [3,8,9,10]. Many of these works could be considered as baselines for comparison.

- I believe the statement in the abstract, "This paper represents an important milestone in generative retrieval research by showing, for the first time, that generative retrieval models can be trained to perform effectively on large-scale standard retrieval benchmarks" (Lines 13-17), is overclaimed. The experiments are conducted on MS MARCO and TREC DL, which have been widely used in previous generative retrieval works. I would agree with the claim if the experiments were conducted on larger datasets, not limited to millions of data.

- Regarding Table 2 in the experiments, it would be beneficial to consider other advanced Product Quantization (PQ) techniques for comparison. There are numerous works building upon PQ, within the IR community as well, that are not discussed in the contribution [11,12].

## Summary:
This paper addresses the key challenges of scaling up generative retrieval models. The proposed techniques and thorough experiments demonstrate that generative retriever can achieve high effectiveness at scale. However, the existing experimental results may insufficiently support the conclusion. Besides, some important discussions are missing.
[1] Zhou, Yujia, Jing Yao, Zhicheng Dou, Ledell Wu, Peitian Zhang, and Ji-Rong Wen. "Ultron: An ultimate retriever on corpus with a model-based indexer." arXiv preprint arXiv:2208.09257.
[2] Sun, Weiwei, Lingyong Yan, Zheng Chen, Shuaiqiang Wang, Haichao Zhu, Pengjie Ren, Zhumin Chen, Dawei Yin, Maarten de Rijke, and Zhaochun Ren. "Learning to Tokenize for Generative Retrieval." NeurIPS 2023.
[3] Li, Yongqi, Nan Yang, Liang Wang, Furu Wei, and Wenjie Li. "Multiview Identifiers Enhanced Generative Retrieval." ACL 2023.
[4] Chen, Jiangui, Ruqing Zhang, Jiafeng Guo, Yixing Fan, and Xueqi Cheng. "GERE: Generative evidence retrieval for fact verification." SIGIR 2022.
[5] Chen, Jiangui, Ruqing Zhang, Jiafeng Guo, Maarten de Rijke, Yiqun Liu, Yixing Fan, and Xueqi Cheng. "A Unified Generative Retriever for Knowledge-Intensive Language Tasks via Prompt Learning." SIGIR 2023.
[6] Wang, Zihan, Yujia Zhou, Yiteng Tu, and Zhicheng Dou. "NOVO: Learnable and Interpretable Document Identifiers for Model-Based IR." CIKM 2023.
[7] Chen, Jiangui, Ruqing Zhang, Jiafeng Guo, Yiqun Liu, Yixing Fan, and Xueqi Cheng. "CorpusBrain: Pre-train a Generative Retrieval Model for Knowledge-Intensive Language Tasks." CIKM 2022.
[8] Ren, Ruiyang, Wayne Xin Zhao, Jing Liu, Hua Wu, Ji-Rong Wen, and Haifeng Wang. "TOME: A Two-stage Approach for Model-based Retrieval." ACL 2023.
[9] Chen, Xiaoyang, Yanjiang Liu, Ben He, Le Sun, and Yingfei Sun. "Understanding Differential Search Index for Text Retrieval." ACL 2023.
[10] Ziems, Noah, Wenhao Yu, Zhihan Zhang, and Meng Jiang. "Large Language Models are Built-in Autoregressive Search Engines." ACL 2023.
[11] Zhang, Han, Hongwei Shen, Yiming Qiu, Yunjiang Jiang, Songlin Wang, Sulong Xu, Yun Xiao, Bo Long, and Wen-Yun Yang. "Joint learning of deep retrieval model and product quantization based embedding index." SIGIR 2021.
[12] Zhan, Jingtao, Jiaxin Mao, Yiqun Liu, Jiafeng Guo, Min Zhang, and Shaoping Ma. "Learning discrete representations via constrained clustering for effective and efficient dense retrieval." WSDs 2022.

**Questions:**

1. Could you provide some details on how to solve the problem of DocID repetition, i.e., multiple documents sharing the same DocID?

2. You demonstrated that residuals quantization (RQ) is more effective for document ID construction than product quantization (PQ). Can you provide some analysis on the causes behind this - does the hierarchical embedding structure explain the difference?

3. You focused on ranking metrics but ultimately usability, latency and throughput are vital. Can generative retrieval achieve competitive speed and resource utilization compared to established sparse/dense retrievers?

4. If two documents share similar topics, it makes sense for them to have similar prefixes. However, in general, a document may contain different topics, could you explain how to deal with this case for the generated prefix of DocIDs?

**Reviewer Confidence:**

4: The reviewer is certain that the evaluation is correct and very familiar with the relevant literature

**Scope:**

4: The work is relevant to the Web and to the track, and is of broad interest to the community

---

### Official Review · Reviewer_21dC · 2023-11-24

**Novelty:** 6
**Technical Quality:** 6

**Review:**

Summary :

The paper introduces RIPOR (Relevance-based Identifiers for Prefix-Oriented Ranking), a framework designed to enhance generative retrieval models by addressing two primary issues inherent in existing models. These models, which use transformer networks as differentiable search indexes, have previously struggled with large-scale, real-world data, limiting their practical application. However, the authors used a prefix-oriented ranking optimization algorithm to overcome some of the current methods' drawbacks. By focusing on prefix-oriented ranking, the proposed strategy aims to reduce noise in the beam search decoding. Additionally, the authors propose quantizing relevance-based representations learned for documents, aligning document IDs more closely with query-document relevance associations. The authors demonstrate that their method shows superior performance compared to other generative IR models and even competitive performance with state-of-the-art dense retrievers.

---------

Strenghts:

1. RIPOR's prefix-oriented ranking optimization and relevance-based document ID construction are novel and address key limitations of existing generative retrieval models. In general, the approach is highly interesting and incorporates several technical innovations.
2. RIPOR significantly outperforms state-of-the-art generative retrieval models on standard retrieval benchmarks.
3. The focus on prefix-oriented optimization specifically targets the challenges posed by beam search in generative retrieval. This idea could potentially be applied to other applications where noise in the initial stages of beam search affects performance.
4. The proposed methodology's components are thoroughly analyzed through detailed ablation studies.
5. The inclusion of a comprehensive set of baselines.
--------------
Weaknesses

1. While the approach is novel and shows promising performance, the complexity and computational cost of the model are not discussed in comparison to other baselines.

2. One concern relates to prefix optimization in scenarios where labels are not sparse. For instance, when there are multiple relevant documents per query, generating training triples with query, positive document, and negative documents may introduce challenges if these documents share a significant portion of prefixes. This could result in labeling paradoxes within the prefixes.

3. The related work section is relatively short. It would be beneficial for the authors to consolidate and provide a more comprehensive review of related work in a dedicated section.

--------------
Comments
- For future venues, I suggest that the authors consider sharing their code for review through https://anonymous.4open.science/. This would facilitate a better understanding of the paper's reproducibility by reviewers.

**Questions:**

1. In section 3.2, why did the authors choose to normalize \alpha_i in that specific manner? Additionally, the rationale behind selecting \beta as 2 is not clear.

2. Please address my concern about potential conflicts in the prefix labels when dealing with comprehensive labels (i.e., more than one relevant document per query). How does RIPOR handle paradoxes in the labels within the prefixes?

3. I wonder if there is any way to analyze the extent to which errors arise from stacking error in beam search or from the last equation of section 3.1 involving previous prefixes.

4. Could you provide information about the training time of RIPOR?

**Ethics Review Description:**

The authors put the paper on Arxiv a few days ago. I am not sure if this is against web conf codes of conduct or not. but I assumed putting the paper on arxiv could have been allowed only before the submission.

**Ethics Review Flag:**

Yes

**Reviewer Confidence:**

3: The reviewer is confident but not certain that the evaluation is correct

**Scope:**

4: The work is relevant to the Web and to the track, and is of broad interest to the community

---

### Official Review · Reviewer_egUH · 2023-11-26

**Novelty:** 4
**Technical Quality:** 5

**Review:**

The paper introduces RIPOR, a novel generation retrieval framework with a prefix-oriented ranking optimization algorithm and a relevance-based document ID construction. The paper claims that RIPOR significantly outperforms existing generative retrieval models on benchmarks like MSMARCO and TREC Deep Learning Track.

### Strengths:
1. The topic is important as generative retrieval has been shown to not work well on big corpora.
2. The writing is clear and easy to follow.
3. The RIPOR achieves significant performance gain against existing generative retrieval methods on big corpora.

### Weakness:
1. The reported results on MSMARCO Dev seem to be incorrect. The reported metrics for baseline models are not consistent with existing paper. For example, ANCE should achieve 33.0 in MRR@10[1], instead of 30.1 reported in the paper; TAS-B should achieve 34.0 in MRR@10[2]. The proposed RIPOR cannot significantly outperform these baselines.
2. Efficiency concern. There are multiple iterations of optimization including 2 iterations in DocID Initialization, 1 in Seq2seq Pre-training, and 3 in Rank-oriented Fine-tuning. Can you provide an amortized time cost for each training iterations, including the time for mining negatives and training with these negatives?
3. Compared with traditional dense retriever, all iterations after DocID Initialization are specific to RIPOR. A direct comparison of $M^0$ (as dense retriever) and RIPOR is desired to show if all the efforts for optimizing the model worth it.
4. The reproducibility is another concern oof the paper. There is no source code  provided, the training recipe looks complicated.

**Questions:**

Questions:
1. What's the teacher $T(q,d^+,d^-)$ in your implementation? Is it a reranker based on docids prefixes?
2. Do you use peudo-queries in Initial Fine-tuning, Prefix-Oriented Rank-oriented Fine-tuning, and Self-Negative Fine-tuning? If so, what is the time cost for mining negatives for these queries in each stage? If not, as there is only 532K training queries but 8.8M passages, how do you guarantee that $M^4$ can produce accurate relevance estimation for unseen docids?

**Reviewer Confidence:**

3: The reviewer is confident but not certain that the evaluation is correct

**Scope:**

3: The work is somewhat relevant to the Web and to the track, and is of narrow interest to a sub-community

---

### Decision · Program_Chairs · 2024-01-22

**Decision:**

Accept (Oral)

**Comment:**

This is a metareview based on the reviews, author responses, and my own opinion. This paper proposes RIPOR, a generative ranker model for large-scale web-search. RIPOR's prefix-oriented ranking optimisation and relevance-based document ID construction are novel and address key limitations of existing generative retrieval models. All of the the referees agree that the work is timely, interesting, and relevant to the community. One key weakness that the authors should address is the oversight of important relevant work in the related work section. While this is very important, I believe that the referees and discussion have given the authors plenty of valuable feedback on how to fix this in the camera ready version of the paper, and I urge the authors to take this seriously. The claim of being "first" to effectively use a generative LLM for ranking is not true, but this does not diminish the novelty of what they have proposed. So, perhaps tone it down a little in the camera ready and focus of the practical impact your approach offers -- which is just as valuable as being "first". This can be addressed with a careful revision of the related work. In general, I like this work and believe it should be accepted. We hope the detailed reviews and discussion will help you produce a fantastic camera ready version for the conference.